# Ovarian Cancer-Associated Mesothelial Cells: Transdifferentiation to Minions of Cancer and Orchestrate Developing Peritoneal Dissemination

**DOI:** 10.3390/cancers13061352

**Published:** 2021-03-17

**Authors:** Kazumasa Mogi, Masato Yoshihara, Shohei Iyoshi, Kazuhisa Kitami, Kaname Uno, Sho Tano, Yoshihiro Koya, Mai Sugiyama, Yoshihiko Yamakita, Akihiro Nawa, Hiroyuki Tomita, Hiroaki Kajiyama

**Affiliations:** 1Department of Obstetrics and Gynecology, Nagoya University Graduate School of Medicine, 65 Tsuruma-cho, Showa-ku, Nagoya 466-8560, Japan; mogi.kazumasa@med.nagoya-u.ac.jp (K.M.); iyoshi.shohei@med.nagoya-u.ac.jp (S.I.); kitami.kazuhisa@med.nagoya-u.ac.jp (K.K.); uno.kaname@med.nagoya-u.ac.jp (K.U.); tano.sho@med.nagoya-u.ac.jp (S.T.); 2Spemann Graduate School of Biology and Medicine, University of Freiburg, Albertstr. 19A, 79104 Freiburg, Germany; 3Division of Clinical Genetics, Lund University, Sölvegatan 19, 22184 Lund, Sweden; 4Bell Research Center, Department of Obstetrics and Gynecology Collaborative Research, Nagoya University Graduate School of Medicine, 65 Tsuruma-cho, Showa-ku, Nagoya 466-8550, Japan; koya.yoshihiro@c.mbox.nagoya-u.ac.jp (Y.K.); mai-sugiyama@kishokai.or.jp (M.S.); yoshihiko-yamakita@kishokai.or.jp (Y.Y.); nawa2005@med.nagoya-u.ac.jp (A.N.); 5Department of Tumor Pathology, Gifu University Graduate School of Medicine, 1-1 Yanagido, Gifu 501-1194, Japan; h_tomita@gifu-u.ac.jp

**Keywords:** ovarian cancer, peritoneal dissemination, mesothelial cells, epithelial mesenchymal transition, tumor microenvironment

## Abstract

**Simple Summary:**

Ovarian cancer has a poor prognosis and tends to develop peritoneal dissemination. It is becoming increasingly clear that mesothelial cells that cover the peritoneal cavity are involved in ovarian cancer progression. Mesothelial cells are numerous in the peritoneal cavity and normally play many functions to maintain the peritoneal environment. In the tumor microenvironment, mesothelial cells that gain mesenchymal features and cooperate with ovarian cancer are called ovarian cancer-associated mesothelial cells, and they must help ovarian cancer cells to adhere to the peritoneum, invade, and disseminate. Elucidating and targeting these processes may improve the prognosis of ovarian cancer, which is difficult to cure.

**Abstract:**

Ovarian cancer has one of the poorest prognoses among carcinomas. Advanced ovarian cancer often develops ascites and peritoneal dissemination, which is one of the poor prognostic factors. From the perspective of the “seed and soil” hypothesis, the intra-abdominal environment is like the soil for the growth of ovarian cancer (OvCa) and mesothelial cells (MCs) line the top layer of this soil. In recent years, various functions of MCs have been reported, including supporting cancer in the OvCa microenvironment. We refer to OvCa-associated MCs (OCAMs) as MCs that are stimulated by OvCa and contribute to its progression. OCAMs promote OvCa cell adhesion to the peritoneum, invasion, and metastasis. Elucidation of these functions may lead to the identification of novel therapeutic targets that can delay OvCa progression, which is difficult to cure.

## 1. Introduction

Ovarian cancer (OvCa) is the primary cause of death among patients with malignant gynecologic neoplasms. The prevalence of OvCa has increased over the past two decades [1,2]. More than half of OvCa patients are diagnosed at an advanced stage, and most patients present with peritoneal dissemination [3,4]. Platinum-based chemotherapy is usually effective for these patients, and approximately 50–70% of patients experience complete or partial tumor remission following first-line treatment [5,6]. However, half of all patients develop recurrent tumors—usually in the peritoneum—even after primary tumor excision [5,6]. Based on these features, controlling peritoneal dissemination is the key to improving the prognosis of OvCa.

According to the “seed and soil” hypothesis [7,8], the peritoneum offers a highly compatible microenvironment for metastatic OvCa cells [9]. However, normally the peritoneum is covered by a hyaluronic acid-based anti-adhesion film and serves as a barrier to external stress [10]. OvCa cells are thought to alter the peritoneum into a favorable environment before creating a peritoneal disseminating site, similar to the cultivation of soil prior to the planting of seeds. In fact, like infection or endometriosis, OvCa causes chronic peritoneal inflammation and disrupts the integrity of the peritoneal environment [11]. Thus, the entire peritoneal ecosystem—including both OvCa cells and the altered host environment—should be regarded as the treatment target rather than focusing on treating the OvCa cells alone.

Histologically, the peritoneal surface is covered by a monolayer of mesothelial cells (MCs) [12], which are thought to play a major role in OvCa-induced peritoneal environment alteration. Studies have reported that OvCa-stimulated MCs gain mesenchymal features and promote peritoneal dissemination of OvCa [13,14,15,16,17,18,19]. Despite these studies, the role of MCs in the peritoneal dissemination of OvCa remains unclear.

In this review, we defined MCs altered by OvCa stimulation as OvCa-associated MCs (OCAMs) and reviewed their role in the peritoneal dissemination of OvCa. We highlight the special function of OCAMs in each phase of metastasis in terms of direct and indirect cell-to-cell crosstalk with OvCa cells.

## 2. Characteristics and Role of MCs and the Association with OvCa

Embryologically, the mesothelium develops from the mesodermal tissue from the 8th to 18th days of gestation, depending on the species. In humans, this occurs around the 14th day of gestation, with cells gradually differentiating from round or cuboidal cells to elongated flattened cells that line the coelomic cavities [20]. The peritoneum covers a large surface area—on average more than 1.0 m^2^ in adults—that is equal to the entire body surface area [21,22]. The peritoneal MCs line the entire peritoneum and are the largest component of the abdominal cavity, with an enormous number of cells. MCs provide a protective surface through the production of phospholipids and a well-developed glycocalyx—a fuzzy coat on the external surface of their plasma membrane consisting of glycolipids and glycoproteins—which allows movement of organs within the serosal cavities [12]. The glycocalyx comprises cell-bound proteoglycans, which consist of a core protein (e.g., syndecan family protein) as well as glycosaminoglycan side chains and sialoproteins [23]. While the primary functions of the mesothelial glycocalyx are known to include maintenance of hydration and minimization of frictional forces in usual visceral maintenance [24,25,26], its role in the dissemination of OvCa is unknown. Mesothelial tissue has been viewed as a sheet of monolayer cells with low mobility, mainly serving as a barrier. However, recent studies have reported that MCs can migrate on the peritoneum and exist as floating cells in ascites fluid [27,28]. Furthermore, MCs have also been reported to contribute to various biological roles such as wound healing and immune function (antigens presenting cells) [29,30]. In this context, MCs may act defensively against tumors. Nevertheless, it has been suggested that they have a tumor-supporting function in the microenvironment in which tumors are developing peritoneal dissemination [31]. Therefore, the role of peritoneal MCs in the tumor microenvironment must be particularly important in advanced-stage OvCa, which is highly likely to develop peritoneal dissemination.

OvCa is the seventh most diagnosed cancer among women worldwide [32]. The mortality rate associated with OvCa is 69%, in contrast to the 19% rate associated with breast cancer. The high mortality rate of this tumor is largely because the majority (75%) of patients are diagnosed at an advanced stage and with widely metastatic disease within the peritoneal cavity [33]. OvCa includes both epithelial and non-epithelial tumors, but 80–90% are epithelial cell tumors [32,33]. OvCa cells are remarkably similar to epithelial cells from extraovarian sites in the female reproductive tract [34]. None of the normal ovarian constituents are lined with epithelial cells resembling those that line the fallopian tube, endometrium, or endocervix [34]. Embryologically, these sites are derived from the Müllerian ducts (paramesonephric ducts) from mesodermal tissues [34]. The fact that peritoneal MCs and OvCa cells are close in origin—derived from the mesoderm—may be one of the possible reasons why OvCa is characterized by a tendency to develop peritoneal dissemination. High-grade serous carcinoma is the most common and most deadly histological subtype of OvCa, and it generally arises from the epithelium in the fimbriated distal part of the fallopian tubes and not from the ovaries [35]. Although the ability to infiltrate the peritoneum varies according to the histological subtype, individual case, and cell lines, we mainly focused on reports related to high-grade serous carcinoma in this review.

MCs are known to present with the specific markers listed in Table 1. Calretinin is a calcium-binding protein of 29 kDa. It is a member of the large family of EF-hand proteins, to which the S-100 protein also belongs. It has been reported that calretinin is highly expressed in MC-lined serosal membranes, and its immunoreactivity is not a feature of common epithelial tumors of the ovary [36], suggesting that it can be useful as a tool to distinguish MCs from OvCa cells. WT1 is a transcription factor expressed in genitourinary tissues. It has been reported that WT1 is also expressed at high levels in many supportive structures of mesodermal origin in mice [37,38]. Podoplanin is a transmembrane mucoprotein recognized by the D2-40 monoclonal antibody [39]. It is strongly and selectively expressed in the lymphatic endothelium. As it is also strongly expressed in normal and neoplastic MCs, podoplanin can be used as an immunohistochemical marker for the diagnosis of both benign and malignant mesothelial lesions [39]. However, podoplanin has also been reported to be present in a small percentage of OvCa cells [40]. Mesothelin is a glycosylphosphatidylinositol-linked cell surface molecule expressed in the mesothelial lining of the body cavities and in many tumor cells [41,42,43,44]. Mesothelin has also been reported to be involved in the adhesion of cancer cells to the mesothelium [43,44]. However, mesothelin has low specificity for MCs, and a more specific mesothelial marker is needed. LRRN4 and UPK3B have been reported as mesothelial markers with higher specificity than mesothelin [45], and further research is underway. In the literature, in pleural effusion cytology desmin is detected in 47 of 56 (84%), N-cadherin is expressed in 48 of 56 (86%), and calretinin is detected in 52 of 56 (93%) of reactive MCs. Moreover, calretinin and cytokeratin 5/6 are reported to be the most promising markers for both benign and malignant MCs [46]. On the other hand, mesothelin, podoplanin, and WT-1 may assist in the diagnosis of MCs. Conversely, some reports have indicated that immunocytochemical probes suitable for the positive identification of MCs and mesotheliomas are scarce because of their limited specificity or sensitivity, or their lack of commercial availability [36,47]. Overall, as it is difficult to identify MCs with complete specificity by immunostaining, appropriate evaluation using a combination of various antigens, morphology, and localization is necessary for the diagnosis of MCs. Therefore, in order to specifically identify MCs in an in vivo experiment, it would be reasonable to use an advanced technology model, such as lineage-tracing [48] or single-cell profiling [49], in addition to the staining described above.

## 3. Biological Process of Peritoneal Dissemination of OvCa

In 1889, Dr. Stephen Paget proposed the original “seed and soil” hypothesis, suggesting that the organ preference patterns of tumor metastasis were the product of favorable interactions between metastatic tumor cells (the “seed”) and their organ microenvironment (the “soil”) [7]. The fact that certain tumors exhibit a predilection for metastasis to specific organs has been recognized by an extensive body of clinical data, and experimental research has confirmed this hypothesis [8]. OvCa shows a characteristic metastatic pathway of peritoneal metastasis via ascites, in addition to the hematogenous and lymphatic pathways. From the viewpoint of the “seed and soil” hypothesis, it is thought that OvCa cells (the “seed”) prefer the peritoneum as the secondary site of growth and development (the “soil”) in the environment of advanced disease.

There are three major processes in the peritoneal dissemination of OvCa (Figure 1). At first, OvCa cells exfoliate as a single cell or a cluster from the primary cancer site in the fallopian tubes or the ovaries and are carried by the physiological movement of ascites to the peritoneum. Subsequently, OvCa cells adhere to the peritoneum and survive on it. Finally, proliferating OvCa cells form tumor nodules with surrounding host cells and maintain the tumor microenvironment.

In the second step—adhesion of OvCa cells on the peritoneum, the first cells that OvCa cells encounter are MCs, located on the peritoneal surface. In the third step—the development of metastatic nodules, MCs are known to present deep in the invasive foci of stroma and support OvCa progression in their role, similar to cancer-associated fibroblasts (CAFs) [13]. CAFs are recognized as a major constituent of the tumor microenvironment, and they play an important role in tumor growth and progression in various cancers, including OvCa [50]. There are various reports on their cells of origin, including fibroblasts, mesenchymal stem cells, bone marrow-derived myofibroblasts, and MCs [50]. MCs act as CAFs by epithelial–mesenchymal transition in the early stage of OvCa metastasis and promote colonization, suggesting that MCs actively contribute to metastasis [51]. Several reports have proposed the renaming of the conversion of MCs that occurs in various organs to mesothelial–mesenchymal transition (MMT), which has been reported to render the peritoneum more receptive to tumor cell adhesion and invasion and contributes to secondary tumor growth by promoting tumor vascularization [13]. On the other hand, recent studies have reported that when OvCa cells exfoliate, they aggregate as spheroids, facilitating adhesion and dissemination [52]. It has also been reported that spheroids contain CAFs as well as tumor cells [52], and MCs may be detected as CAFs in ascites. Based on these pathophysiological features, we believe that MCs can play a vital role in promoting peritoneal dissemination of OvCa.

During peritoneal dissemination, MCs are known to be altered by stimulation from OvCa. The stimulation of soluble factors initiates this conversion in malignant ascites, such as transforming growth factor-β (TGF-β), hepatocyte growth factor, and interleukin (IL)-1β [13,16,53,54]. Additionally, IL-1, IL-6, IL-10, tumor necrosis factor-alpha, and TGF-β are also elevated in malignant ascites of OvCa [55,56]. Among these, TGF-β1 primarily contributes to the alteration, mainly mesenchymal transition [16]. In the field of nephrology, peritoneal fibrosis is recognized as the main cause of the reduction in the effectiveness of peritoneal dialysis [57]. Fibrosis fundamentally stems from the mesenchymal transition of MCs caused by TGF-β1 stimulation, derived from persistent inflammation with continuous peritoneal dialysis [58,59].

In patients with advanced OvCa, the same chronic inflammation and peritoneal fibrosis seem to persist due to cancer promotion in the peritoneum. Additionally, reports have indicated that these cancer-associated MCs promote OvCa progression via various cell-to-cell crosstalk with OvCa cells [9,16,51]. In this context, we comprehensively defined the MCs altered by OvCa as OCAMs and demonstrated their features [9,15,16,17,19]. Based on Dr. Stephen Paget’s theory, seeds of OvCa cells cultivate the soil of MCs for OCAMs of the peritoneum by nourishing them with TGF-β1, which promotes the progression of peritoneal dissemination. In other words, this remarkable biological process of peritoneal dissemination of OvCa can be considered a future therapeutic target for controlling OvCa progression.

## 4. OCAMs Affect OvCa Adhesion on the Peritoneum

### 4.1. Adhesion Molecules

As previously described, OvCa cells shed from the primary tumor through the ascites that land on MCs. Previous studies have reported that OCAMs express various molecules related to OvCa cell adhesion. The representative molecules are listed in Table 2.

First, the extracellular matrix (ECM) plays an essential role in OvCa adhesion to MCs. OCAMs are known to remodel ECM dramatically and promote this process [20,60,61]. The principal components of ECM on MCs are fibronectin, vitronectin, laminin, collagen I, and collagen IV [62]. For example, fibronectin, which is highly elevated in OCAMs, is reported to promote OvCa cell adhesion, which may be a novel therapeutic target [16,51]. On the other hand, it has been reported that vitronectin mediates the adhesion of OvCa cells to the mesothelium in cooperation with its αv integrin and uPAR receptor [63].

Hyaluronan, a high molecular weight glycosaminoglycan present in the ECM, is also produced by MCs [10,64]. CD44 is a broadly distributed cell surface protein that mediates cell attachment to ECM components or specific cell surface ligands [65] and is found on OvCa cells. CD44 binds to the ECM glycosaminoglycan hyaluronan with high affinity and also has a weak affinity for fibronectin, laminin, and collagen types I and IV [65,66,67]. Several reports have suggested that the interaction between CD44 and hyaluronan modulates the adhesion of OvCa cells to the mesothelium [62,64,67].

The immunoglobulin superfamily is a large group of cell adhesion proteins, which include vascular cell adhesion molecule 1 (VCAM-1) and L1 cell adhesion molecule (L1CAM) [60,61]. VCAM-1 is a cell surface receptor expressed on MCs, and it has been suggested that VCAM-1 and its ligand, α4β1 integrin, are involved in OvCa adhesion. L1CAM is a 200–220 kDa transmembrane glycoprotein composed of six Ig-like domains and five fibronectin type III repeats [68]. L1CAM mediates adhesion to the peritoneum by interacting with mesothelial neuropilin-1 [69]. It has also been reported that L1CAM is involved in the regulation of proliferation, migration, and invasion, as well as adhesion of cancer cells [70].

The mucin MUC16, which carries the peptide epitope CA125, is a very large mucin with an average molecular weight of 2.5–5 million Da. It has been reported that mesothelin is a CA125-binding protein and that CA125 might contribute to the metastasis of OvCa to the peritoneum by initiating cell attachment to the mesothelial epithelium via binding to mesothelin [43,44].

Selectins (E-, P-, and L-selectin) are a family of calcium-dependent glycoproteins that facilitate the early stages of the adhesion cascade [71,72]. In particular, the interaction between P-selectin and sialyl-Lewis^x^ is now thought to be important for the adhesion of OvCa to the mesothelium [71,72,73].

Chemokines may also act indirectly in the adhesion of cancer cells to MCs. It has been reported that adhesion of ovarian carcinoma cells to human peritoneal MCs is dependent on CX3CL1/CX3CR1 signaling [74]. Moreover, the SDF-1a/CXCR4 axis has been reported to be involved in adhesion between OvCa cell lines and HPMCs or ECM components [15]. We have also previously reported that CCL2 secreted from OCAM promotes adhesion, migration, and invasion of OvCa with its receptor, CCR2 [75].

Lysophosphatidic acid (LPA) is one of the notable molecules involved in tumor progression, including adhesion. LPA is a ubiquitous phospholipid with growth factor-like activities that acts through a subfamily of G-protein coupled cell surface LPA-specific receptors. In particular, it has been reported that ascites from OvCa patients contain high levels of LPA [76,77]. The production of LPA by both peritoneal MCs and OvCa cells has been shown to promote the metastatic phenotype in OvCa [78]. LPA has also been reported to stimulate adhesion, migration, and invasion of OvCa cells in vitro, and may play similar roles in vivo [79].

### 4.2. OCAMs and OvCa Adhesion

According to the “seed and soil” hypothesis, cell adhesion corresponds to the phenomenon of sprouting in suitable soil, which can be described as a key–hole relationship. In other words, cell adhesion can be established by combining suitable adhesion molecules, leading to the subsequent formation of metastases that may be because they are closely related in terms of embryology. The changes in OCAMs are mainly due to mesenchymal transition but may also be viewed as transdifferentiation. While there are no unique markers for OCAM, it has been reported that the expression of genes encoding proteins that are known markers of epithelial–mesenchymal transition (EMT), such as E-cadherin decrease and N-cadherin increase, is altered in MCs induced by TGF-β1 [9]. We speculate that as cancer cells become more malignant as they become more undifferentiated, they may become more closely related to each other as they transdifferentiate. It is essential to inhibit the mesenchymal transition or the transdifferentiation of host cells to prevent this, which may be a future therapeutic target.

Moreover, OvCa cells that have adhered to mesothelial tissue are thought to subsequently infiltrate the sub-mesothelium through the intercellular spaces between MCs. Mesothelial clearance is recognized as a phenomenon in which MCs are displaced from the adhesion site [80]. It has been reported that MCs loss is observed on the peritoneal surface very early in the adhesion of OvCa cells, and this is one mechanism that has been proposed [51,81]. However, it is also possible that the MCs actively provide the defective area to the cancer cells, while this event can be regarded as a passive detachment of MCs under the influence of OvCa cells. The reason for this is that MCs have been shown to move horizontally in both in vitro and in vivo experiments [27]. The cobblestone-like surface structure is a dynamic structure, rather than a barrier-like structure that remains in place. As MCs can migrate to the peritoneum, we speculate that the adhesion of OvCa cells may trigger MCs to expand the cellular gap by their own migration, inviting cancer cells into the sub-mesothelial layer. Experimentally, this phenomenon is referred to as trans-mesothelial migration of OvCa cells (Figure 2). In our previous report, the filopodia of OvCa cells was associated with a significant role in trans-mesothelial migration [82]. On the contrary, the active migration of MCs affected by OvCa cells is still unclear. A multi-perspective evaluation of this event may lead to the elucidation of new mechanisms and the development of novel adhesion-preventive therapies targeting host cells.

## 5. OCAMs Represent One of the Stromal Components Promoting Tumor Microenvironment of Peritoneal Dissemination

CAFs are derived from various cells, and originate from different sources, depending on each tumor microenvironment [83]. As the peritoneal cavity is the main stage of OvCa, abundant MCs in the peritoneal cavity can be one of the origins of CAFs. It has been reported that in peritoneal metastasis, a subpopulation of CAFs originates from MCs through a mesenchymal transition, predominantly initiated by TGF-β1 [13,14,16]. In our previous report, histological analysis of peritoneal dissemination of OvCa revealed the presence of myofibroblasts, consistently associated with peritoneal MCs, suggesting that OCAMs are able to invade the peritoneal tissues together with OvCa cells [9]. In gastric cancer experiments, cancer-activated MCs were also found to create an invasion front by guiding cancer cells [84]. These findings suggest that during the development of peritoneal dissemination, mesenchymally transited OCAMs and transdifferentiating MCs possibly invade submesothelial tissue and create a tumor microenvironment with OvCa cells.

As OvCa cells and OCAMs co-exist in the tumor microenvironment of peritoneal dissemination, it is reasonable to hypothesize that both cells mutually interact via cell-to-cell crosstalk and are involved in the development of further cancer promotion. However, few reports have studied the impact of OCAMs on OvCa cells in the tumor microenvironment of the peritoneal dissemination of OvCa. We previously revealed a partial effect of OCAMs in the stroma of peritoneal dissemination using an in vitro co-culture model with OvCa cells and OCAMs. However, there were some experimental limitations, such as the difficulty in identifying and tracing OCAMs in vivo, and the real existence of OCAMs in tumoral stroma has not yet been proven [9]. On the other hand, stromal heterogeneity of CAFs has been reported in some types of cancers, including primary OvCa [85]. Moreover, MCs are reported as a partial component of stromal cells in pancreatic cancer [49]. Altogether, we can speculate that the transdifferentiating MCs supply stromal resources and contribute to the creation of the tumor microenvironment.

Although the definite function of OCAMs in the stroma of peritoneal dissemination is not well understood, it may play a key role in controlling the progression of OvCa. We previously identified that OvCa cells acquire platinum resistance via activation of the Akt signaling pathway, which is induced by fibronectin on the surface of OCAMs [9]. The Akt signaling pathway is known to be associated with platinum resistance in OvCa cells [86,87,88,89]. In addition, fibronectin has been reported to play a critical role in the progression of OvCa cells to the peritoneum and also induces resistance to some chemotherapeutic agents [90,91,92]. Therefore, if we can inhibit the conversion of MCs to OCAMs, it may be possible to effectively control the progression of peritoneal dissemination by mediating the pathways related to the refractory disease.

## 6. Future Prospects

The clinical outcomes of patients with refractory OvCa have not improved significantly over the past few decades [93]. In particular, the prognosis of OvCa with peritoneal dissemination is still poor, and there is a demand for new therapeutic strategies to control tumor progression.

The success of bevacizumab—which inhibits the effect of vascular endothelial growth factor by targeting neoplastic angiogenesis—in significantly prolonging progression-free survival and showing a synergistic effect on platinum agents, has shed light on tumor microenvironment-targeted therapy strategies in OvCa [93]. As we have explained above, through stimulation from OvCa cells, MCs can be transdifferentiated to create a pro-tumorigenic milieu that orchestrates the development of peritoneal dissemination. This indicates that each process of mesothelial conversion can be a therapeutic target that controls the progression of OvCa. For instance, when we focus on the chemokine signal in the adhesion process, the SDF1-CXCR4 axis is a druggable target, and preclinical trials using blocking reagents are underway on other malignancies, including pancreatic ductal adenocarcinoma [94]. Inhibiting the altered activation of OCAMs, ATRA (Vitamin A analog), or Calcipotriol (Vitamin D analog) may possibly induce cancer-suppressing effects as seen in CAF-targeting cancer therapy trials [95,96,97]. As mentioned above, the reported surface markers for MCs still lack high sensitivity and specificity; however, once such ideal markers for activated MCs are established, even a direct elimination of these cells would be a strategic option by applying oncolytic adenovirus or transgenic technologies [98,99]. In addition, as we described, it has been reported that peritoneal MCs form part of the cancer-associated fibroblasts in pancreatic cancer stroma [49] and some in OvCa peritoneal dissemination [13]. Therefore, targeting MCs and searching for novel therapeutic candidates should be beneficial not only for advanced OvCa but also for other types of cancer with peritoneal dissemination.

These strategies do not directly damage cancer cells but are expected to suppress OvCa cells indirectly by normalizing the tumor microenvironment. At a glance, this may seem like a detour, but it must form one of the effective treatment strategies for OvCa with presumably fewer side effects and possible synergetic effects in addition to conventional chemotherapy regimens. Complete eradication of cancer cells would be impossible in advanced OvCa as occult metastasis remains even in patients who experience initial complete remission of the tumor [100]. For this reason, it is clear that advanced OvCa is a chronic disease that remains in the host body. Based on this, we believe that future OvCa therapeutic strategies should focus on facilitating the longest possible tumor-bearing life expectancy for patients. Further basic and clinical research is needed to develop breakthrough treatments that provide patients with an improved quality of life following OvCa.

## Figures and Tables

**Figure 1 cancers-13-01352-f001:**
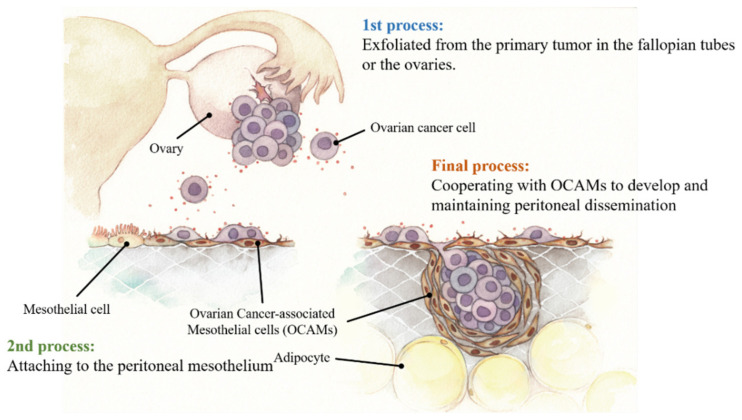
Three processes of development of peritoneal dissemination from ovarian cancer (OvCa). First, OvCa cells are exfoliated from the primary tumor in the fallopian tubes or the ovaries. Second, the free OvCa cells encounter the peritoneal mesothelium and attach to it. Finally, OvCa cells cooperate with OvCa-associated mesothelial cells (OCAMs) to develop and maintain peritoneal dissemination.

**Figure 2 cancers-13-01352-f002:**
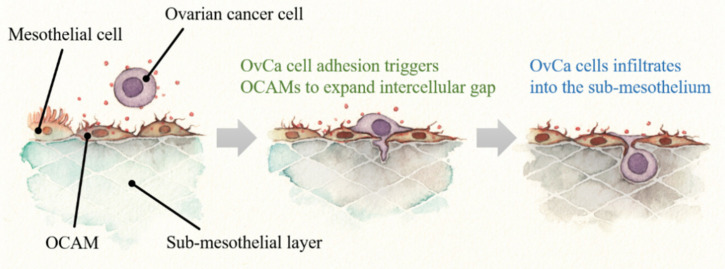
The model of trans-mesothelial migration of ovarian cancer (OvCa) cells. OvCa cells stimulate mesothelial cells (MCs) to transition into OvCa-associated MCs (OCAMs), which in turn enhance the adhesion of OvCa cells to mesothelial tissue. This adhesion triggers OCAMs to expand the intercellular gap by their own migration. Then, OCAMs invite the OvCa cells to pass through the intercellular gap and enter the sub-mesothelial space.

**Table 1 cancers-13-01352-t001:** Mesothelial markers.

Markers	Location	Articles
Calretinin	Intracellular, Cytosol	[36]
Podoplanin	Plasma membrane	[39,40]
WT1	Intracellular, Nucleoplasm	[37,38]
Mesothelin	Intracellular, Vesicles and Nucleoplasm	[41,42,43,44]
Desmin	Intracellular, Intermediate filaments	[46]
Cytokeratin 5/6	Intracellular, Intermediate filaments	[46]
N-cadherin	Plasma membrane	[46]
LRRN4	Intracellular, Nucleoli	[45]
UPK3B	Plasma membrane, Intracellular, Cytosol	[45]

**Table 2 cancers-13-01352-t002:** Adhesion molecules.

Ligands		Receptors	References
ECM			
	Laminin, fibronectin, vitronectin, collagen I, and IV	Integrins	[16,20,51,60,61]
	Vitronectin	uPAR	[63]
	Hyaluronan	CD44	[62,64,65,66,67]
Membrane protein			
	VCAM1	Integrins	[60,61,68]
	Neuropilin-1	L1CAM	[60,61,69,70]
	Mesothelin	MUC16	[43,44]
	P-selectin	sialyl-Lewis^x^ (sLe^x^)CD24	[71,72,73]
Chemokine			
	SDF-1	CXCR4	[15]
	CX3CL1	CXCR1	[74]
	CCL2	CCR2	[75]

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
