# Peer review of "Ovarian Cancer-Associated Mesothelial Cells: Transdifferentiation to Minions of Cancer and Orchestrate Developing Peritoneal Dissemination"

_cancers, 2021, doi:10.3390/cancers13061352_

Round 1
Reviewer 1 Report
In this review, the authors have described the role of mesothelial cells in promoting ovarian cancer metastasis. They have very nicely explained the process of peritoneal dissemination. The factors involved in the adhesion of cancer cells to the mesothelial cells are well described. The role of ovarian cancer-associated mesothelial cells (OVCAMs) in the tumor microenvironment is well explained. This review is highly relevant to the field. However, there are a few things if explained a bit more will greatly enhance the reader’s knowledge and pleasure.
- Line 285: please explain a bit more about the results from ref 76.
- Explain transdifferentiation a bit more by elaborating the information from the references cited. Line 242-244 can be explained more, as this interesting.
- Similarly, line 293-294 talks about an in vitro co-culture. It will be better for the reader to know more about it. Then line 295-296 talk about limitations. It is not clear.
- Line 125, please avoid statements like “further report expected in the future”
- Figure 1: Final process, shows a sheath of OVCAMs around the cancer cells. OVCAMs can be interspersed with the cancer cells.
- Authors have clearly explained about CAFs, A few statements on mesothelial to mesenchymal transition in cancer in this segment will be helpful.
Author Response
Dear Reviewer
Thank you very much for your comments on this manuscript.
We have reflected as much as we could.
The following is a list of improvements.
Comment 1. Line 285: please explain a bit more about the results from ref 76.
Answer: The text has been changed to provide more detail. (Lines 306-307)
Comment 2. Explain transdifferentiation a bit more by elaborating the information from the references cited. Line 242-244 can be explained more, as this interesting.
Answer: we added a sentence about molecular differences between MC and OCAM. (Lines 262-265) If possible, I would like to change this content further.
Comment 4. Line 125, please avoid statements like “further report expected in the future”
Answer: The text “in the future” has been deleted. (Lines 247-254)
Comment 5. Figure 1: Final process, shows a sheath of OVCAMs around the cancer cells. OVCAMs can be interspersed with the cancer cells
Answer: As you pointed out, OVCAMs can be interspersed with the cancer cells in the Figure 1, final process. But, it is very sorry that it is difficult to change the figure. We can add supplementary information in the text instead.
Comment 3 and 6 have not been revised yet, but we hope to revise them if we have enough time. We also plan to submit this manuscript for English editing.
I would appreciate your comments on this revised manuscript.
Sincerely,
Reviewer 2 Report
The authors describe in the review manuscript the roles cancer-associated mesothelial cells play in tumor progression and establishment of peritoneal metastasis in ovarian cancer.
The subject for the review is timely and awaited, but I have identified some major errors that must be corrected before the manuscript can be recommended for publication.
I will suggest to write out in full OvCa, MC, OCAM and other custom-made abbreviations.
The statement «Nearly half of them experience complete tumor remission in the first-line treatment» is incorrect. - The number is much higher. This must be corrected.
Ovarian cancer is a term that refers to a heterogeneous group of malignancies arising from or involving the ovaries. Morphologically ovarian cancer is classified into two broad categories: non-epithelial ovarian cancer and epithelial ovarian cancer. The vast majority, 85-90%, belongs to the and epithelial ovarian cancer. Epithelial ovarian cancer can arise from both from ovary fallopian tube or peritoneum as well as the ovaries. Epithelial ovarian cancer can be subdivided majorly into at least five different histological subtypes with different etiologies, genetic backgrounds, phenotypic characteristics, and clinical features. The most common and most deadly histological subtype is high-grade serous ovarian carcinoma. Most of the data presented seems to me to be derived from studies focusing om high-grade serous ovarian carcinoma. The authors must specify this better.
High-grade serous ovarian carcinoma commonly arises from the epithelium in the fimbriated distal part of the fallopian tubes and not form the ovaries. This must also be changed in the text.
The statement «Another reason is that OvCa has a relatively thin capsule, which makes it easier for OvCa cells to physically spread into the peritoneal cavity». – This is not correct for humans. This must be corrected.
Line 137. Lineage-tracing
I will suggest that more approaches like single cell profiling to be included here
TGF-b1 and not TGF-b1 (throughout the manuscript)
Lines 176-178.
The peritoneal cavity in ovarian cancer represents an inflammatory compartment and the malignant ascitic fluid is an exudate. This in contradiction top the fluid generated in less inflammatory conditions. I am unsure if the inflammation described for peritoneal fibrosis due to dialysis is comparable.
Line 197. The word produce should be replaced with expressed.
Paragraph 4 must be shorter.
It is my experience from in vitro experiments that the ability for different cell lines to infiltrate the peritoneum differ. The same is also the case in real life.
The variation should also be high-lighted.
Paragraph 6. Future prospection – Correct word?
I think the authors should present this paragraph differently:
Please select 3 or 4 the main subjects presented earlier. The focus should be more on the biological relevance and in addition to the clinical consequences. A focus on adhesion, EMT and angiogenesis is needed.
Author Response
Dear Reviewer
Thank you very much for your comments on this manuscript.
We have reflected your comments as much as we could.
The following is a list of improvements.
- I will suggest to write out in full OvCa, MC, OCAM and other custom-made abbreviations .
Answer: We added an abbreviation section. (Lines 40-43)
- The statement «Nearly half of them experience complete tumor remission in the first-line treatment» is incorrect. - The number is much higher. This must be corrected .
Answer: Tumor remission rate for first-line therapy has been changed. (Lines 50-51)
- Ovarian cancer is a term that refers to a heterogeneous group of malignancies arising from or involving the ovaries. Morphologically ovarian cancer is classified into two broad categories: non-epithelial ovarian cancer and epithelial ovarian cancer. The vast majority, 85-90%, belongs to the and epithelial ovarian cancer. Epithelial ovarian cancer can arise from both from ovary fallopian tube or peritoneum as well as the ovaries. Epithelial ovarian cancer can be subdivided majorly into at least five different histological subtypes with different etiologies, genetic backgrounds, phenotypic characteristics, and clinical features. The most common and most deadly histological subtype is high-grade serous ovarian carcinoma. Most of the data presented seems to me to be derived from studies focusing om high-grade serous ovarian carcinoma. The authors must specify this better.
Answer: We added a sentence about High-grade serous ovarian carcinoma. I would like to add some more details. (Lines 109-111)
- High-grade serous ovarian carcinoma commonly arises from the epithelium in the fimbriated distal part of the fallopian tubes and not form the ovaries. This must also be changed in the text.
Answer: I’m sorry that these sentences have not been revised yet.
- The statement «Another reason is that OvCa has a relatively thin capsule, which makes it easier for OvCa cells to physically spread into the peritoneal cavity». – This is not correct for humans. This must be corrected.
Answer: The statement «Another reason is that OvCa has a relatively thin capsule, which makes it easier for OvCa cells to physically spread into the peritoneal cavity». – This statement is not correct, and has been deleted.
- Line 137. Lineage-tracing, I will suggest that more approaches like single cell profiling to be included here.
Answer: We added a sentence about single cell profiling (Line 143). I would like to add some more details.
- TGF-b1 and not TGF-b1 (throughout the manuscript)
Answer: TGF-b1 has been replaced with TGF-β1.
- Lines 176-178. The peritoneal cavity in ovarian cancer represents an inflammatory compartment and the malignant ascitic fluid is an exudate. This in contradiction top the fluid generated in less inflammatory conditions. I am unsure if the inflammation described for peritoneal fibrosis due to dialysis is comparable.
Answer: I’m sorry that these sentences have not been revised yet.
- Line 197. The word produce should be replaced with expressed.
Answer: I've changed the word to ”expressed”.
- Paragraph 4 must be shorter.
Answer: Paragraph 4 is long, so we decided to divide it into shorter subsections.
- It is my experience from in vitro experiments that the ability for different cell lines to infiltrate the peritoneum differ. The same is also the case in real life.
The variation should also be high-lighted.
Answer: I’m sorry that these sentences have not been revised yet.
- Paragraph 6. Future prospection – Correct word?
I think the authors should present this paragraph differently:
Please select 3 or 4 the main subjects presented earlier. The focus should be more on the biological relevance and in addition to the clinical consequences. A focus on adhesion, EMT and angiogenesis is needed.
Answer: Paragraph 6 has been changed to show more specific targets.
There are some parts that have not been revised yet, but we hope to revise them.
We also plan to submit it for English editing.
I would appreciate your comments on this revised manuscript.
Sincerely,
Reviewer 3 Report
The review manuscript entitled “ Ovarian cancer-associated mesothelial cells: Transdifferentiation to minions of cancer and orchestrate developing peritoneal 3 dissemination” by Kazumasa Mogi et al has focused on an important aspect of ovarian cancer. While it covers the progression in the field, it needs some major revision to enhance the quality of the review.
- The logic flow and organization of the review need to be significantly improved.
- Section 4 needs to be divided into subsections and significantly expanded to cover the major role and molecular mechanisms of each interactive molecule.
- The clear molecular differences between MC and OCAM should be better described.
- In particular, whether the interactions are through direct contact or secreted molecules should be reviewed.
- The authors should mention an important interactive molecule between MC and EOC, LPA (e.g. PMIDs: 17325741, 16540649, and 16171822).
- The in vitro (included 3D) and in vivo model systems used for the interactions should be reviewed.
- Lines 151, 196, and Fig. 1, the authors mentioned, “At first, OvCa cells exfoliate as a single cell or a cluster from the primary cancer site ….”. Ovarian cancer cells may be shed from the surfaces of fallopian tubes or ovaries, which is a very important concept and needs to be incorporated.
- The Conclusion section should be revised to remove some loosely relevant contents (such as the VEGF part can be shorten to 1-2 sentences) and add more perspective on which one to target and how to target the interactions.
- English needs to be modestly edited. Many of the connection words are improperly used, including, but not limited to, Line 158: Moreover; Line 180: However; line 318, In contrast etc.
Minor:
- TGF-b1 in the text should be changed to TGF-β1.
Author Response
Dear Reviewer
Thank you very much for your comments on this manuscript.
We have reflected as much as we could.
The following is a list of improvements.
Comment 2. Section 4 needs to be divided into subsections and significantly expanded to cover the major role and molecular mechanisms of each interactive molecule.
Answer: Paragraph 4 is long, so we decided to divide it into shorter subsections. I would like to add more details.
Comment 3. The clear molecular differences between MC and OCAM should be better described.
Answer: We added a sentence about molecular differences between MC and OCAM. (Lines 262-265 )
Comment 5. The authors should mention an important interactive molecule between MC and EOC, LPA.
Answer: We added sentences about LPA. (lines 247-254)
Comment 8. The Conclusion section should be revised to remove some loosely relevant contents (such as the VEGF part can be shorten to 1-2 sentences) and add more perspective on which one to target and how to target the interactions.
Answer: section 6 has been changed to show more specific targets.
I’m sorry that comment 1, 4, 6, 7 have not been revised yet, but we hope to revise them.
We also plan to submit it for English editing.
I would appreciate your comments on this revised manuscript.
Sincerely,
Round 2
Reviewer 2 Report
The manuscript has improved much.
The subjects I raised have been properly answered.
The manuscript still needs a proper English editing, but the scientific content is sufficient.